# Brainstem Glioma Prognostication: Static FET PET/CT

**DOI:** 10.3390/cancers17183065

**Published:** 2025-09-19

**Authors:** Dávid Gergő Nagy, Júlia Singer, Katalin Borbély

**Affiliations:** 1Department of Neurosurgery and Neurointervention, Semmelweis University, 1145 Budapest, Hungary; 2Accelsiors Ltd., 1222 Budapest, Hungary; 3Clinic for Neurology, Semmelweis University, 1083 Budapest, Hungary

**Keywords:** FET PET/CT, brainstem glioma, TBR cutoff value, amino acid PET/CT

## Abstract

Adult brainstem glioma is a rare and poorly understood disease. Functional PET metabolic pattern mapping is gaining increasing attention in the appropriate assessment of brain tumors, adding relevant functional information to conventional and modern MRI sequences. Amino acid PET/CT successfully assists in the oncological treatment of supratentorial gliomas. Despite all this, only a few studies are available on amino acid PET/CT in brainstem gliomas. The aim of our single-center retrospective analysis was to evaluate the diagnostic performance of static amino acid FET PET/CT in ambiguous brainstem lesions. We focused on prognostic capabilities; therefore, the aim of our analysis was to separate short-term and long-term survival subgroups. Our analysis suggests that a higher tumor-to- brain ratio (TBR = 2.9) may be a good cutoff value for distinguishing patients with poor and better prognosis.

## 1. Introduction

The management of patients with brainstem glioma (BSG) is demanding. BSG only consists less than 2% of all glial tumors [1]. The tumor location rarely allows a safe biopsy. Therefore, histological features and molecular patterns are less comprehensively identified than for their supratentorial counterparts. Despite several advancement in diagnostic possibilities valid metabolic mapping is rare. The only proven treatment option is fractionated radiotherapy [2]. Despite all these disadvantages long-term prognosis can be much better than for hemispherical glioblastomas [2,3].

Most of our current knowledge about brainstem gliomas come from MRI morphologic features. Based on the regular sequences four distinct subgroups can be identified.

The most frequent subtype is diffuse intrinsic low grade glioma (~50%; Figure 1) followed by enhancing high grade glioma (Figure 2), and then the focal tectal and exophytic gliomas [3,4,5]. These latter subtypes can have very different prognosis [2]. However biological behavior and MRI morphology not always coincide. The recent fifth edition of WHO CNS tumor classification ranks molecular qualities prior to classic histologic features of tumors [6]. Novel results with targeted oncological therapy possibilities also emphasize the importance of genetic profiling of a tumor [7]. However, heterogeneity of glial tumors was proven several times [8,9]. Certain genetic features can be used as prognostic factor such as IDH mutation for supratentorial gliomas. For brainstem gliomas H3K27M mutation represents a similar role [10,11].

These special conditions lead to further need for advanced imaging in the diagnostic work up. Magnetic resonance spectroscopy (MRS) can assess the levels of specific metabolites or neurotransmitters in defined areas of the brain. These metabolites can depict information about IDH and H3K27M mutation status of a brainstem lesion [12]. Radiomic analysis of certain sequences can also help identify these mutations [13,14] suggesting prognostic predictions. MR spectroscopy can have several limiting factors such as spatial resolution, which is a crucial issue in the compact sized brainstem. Further technical difficulties arise due to the proximity of bony structures of the posterior fossa, both leading to limited number of available cases until now [15,16,17]. FDG PET/CT is the most used metabolic imaging modality in general oncology but due to the high physiological cortical glucose uptake its use in brain tumors is highly limited [18]. Amino-acid (AA) PET/CT has already been proven to be a useful adjunct in the management of supratentorial gliomas in both newly diagnosed and pre-treated cases [19,20,21,22]. Infratentorial application of AA PET/CT lacks the experience and widespread usage. The possible radiolabeled ligands which can be used are C-MET, F-FET, F-DOPA. Based on the practical and diagnostical reasons F-FET is the most widely used tracer.

Amino acid PET CT is evaluated based on tumor brain ratios (TBR) which are calculated from the standardized uptake value (SUV) of the region of interest and healthy part of the brain (consisting grey and white matter) [23]. Maximum and mean TBR are calculated. In dynamic scans a time activity curve is a further aid to describe the lesion [24]. In supratentorial setting 2.5 ≥ TBRmax and 1.9 ≥ TBRmean are widely accepted cutoff values [25] (in previously treated cases TBRmax ≥ 2.3 and TBRmean ≥ 2.0 is also used). Also 1.6 ≤ TBR area is considered neoplastic tissue. Based on these values true biological tumor volume can be calculated.

In our single center retrospective study, we aimed to identify further new aspects of using FET PET/CT for suspected infratentorial gliomas.

## 2. Materials and Methods

As a routine imaging method, we have been using ^18^F-FET PET/CT since November 2019. With our on-site scanner (GE Discovery MI 4 2017 edition, Waukesha, WI, USA) we were able to perform over 500 studies between November 2019 and April 2023. All patient referrals came from physicians treating intrinsic brain tumors. 25 examinations were done for infratentorial gliomas. In order to have a homogenous group multiple scanned patients were only taken account with their initial scan. Our final number of patient eligible for analysis was 20. All patients underwent the same examination protocol. 4 h fasting prior to the intravenous administration of 200 MBq ^18^F-FET with 20-min uptake, followed by 10-min static acquisition. Patients underwent non contrast, normal dose CT and subsequent PET scan in the same supine position. For image reconstruction OSEM and BPL methods were used. Maximum of standardized uptake values (SUVmax) of region of interest and of healthy supratentorial white matter tissue were measured by using InterView™ FUSION 3.08 (MEDISO Ltd., Budapest, Hungary) multi-modality medical image visualization and post-processing software. Subsequently, maximal tumor-to-background ratios (TBRmax = SUVmax target/SUVmax background) were calculated. Images were evaluated by a nuclear medicine specialist and a neurosurgeon as well. In the supratentorial setting our default interpretation for TBRmax over 1.6 is considered to be ^18^F-FET PET positive and possibly malignant lesion. In the infratentorial setting this cutoff produced some conflicting results. Literature can also be inconsistent on what cutoff values should be used for true validation of malignancy and tumor delineation. In the near future radiomic analysis can be a helpful aid to identify these qualities [26]. Our main goal was to identify the best possible TBR value that could reflect on tumor prognosis.

## 3. Results

20 patients were included in this analysis. Female:male ratio was 8:12. Median age was 46 years (27–76 years). MRI contrast enhancement was visible in 12 cases, 8 lesions showed only T2/FLAIR hyperintinsity. 16 cases were newly diagnosed, and 4 patients were previously treated but these scans were their first PET scan. All lesions were correctly identified as neoplastic. 3 patients underwent subsequent sampling. Two lesions were identified as glioblastoma (gr 4) and one as pilocytic astrocytoma (gr 1).

Descriptive statistics for TBR values was prepared overall and by subgroups for deceased and living patients (Table 1). Tumor-brain ratio values were then dichotomized into “Low” and “High” subgroups with different cutoff values in order to obtain the best cut-off in terms of Youden index (unweighted). Sensitivity was defined as the proportion of subjects in the “Low” TBR subgroup among survivors and specificity was defined as the proportion of subjects in the “High” TBR subgroup among death cases. For survival analysis the date of first MRI was used as a starting timepoint and patients still alive at the end of the observation period were censored at the date of last observation. Sensitivity and specificity values were estimated along with their 95% CI. The corresponding negative and positive predictive values were also estimated. Descriptive statistics for the two subgroups (survivors and deceased) were presented. Survival times across the two strata (dichotomized based on the optimal cutoff) were compared by a log-rank test, and the Kaplan-Meier graph for the survival distributions in the two TBR subgroups was also presented. All calculations were performed with SAS^®^ version 9.4. The maximum of the Youden index was attained at TBR = 2.9. Therefore measures for this cutoff are presented separately. The performance characteristics of the cutoff TBR = 2.9 showed good results for sensitivity, and positive and negative predictive value (point estimates of 91.7; 68.8 and 75%, respectively) (Table 2). However specificity is failed to meet the desired levels in our dataset (37.5%).

The log-rank homogeneity test across the two strata corresponding to the two TBR categories yielded a statistically significant result, *p* = 0.026. This was also reflected in the survival distributions in the two strata (Figure 3). The median survival time in the high TBR subgroup was of 199 days (with a 95% lower CI limit of 19 days, the upper limit was non-estimable). The median survival time in the low TBR subgroup was of 1293 days (95% CI limits: 777 to 1293 days).

## 4. Discussion

Brainstem gliomas can have enormous impact on quality of life and the belated treatment may lead to further deterioration of the neurological performance of these patients. Neurosurgical interverventions are so far equivocal with no clear benefit, however, a representative tumor sample can lay out the genetic map of the tumor [27]. The relatively high complication rate due to the eloquent location is restraining the widespread use of surgical biopsy.

Upon our study only a handful of prognostic factors have been statistically proven [28]. For a better understanding the biological behavior of brainstem glioma new indicators, qualities are needed, without invasive sampling.

Solely MRI derived decision making has proven to be outdated [29]. Optimal diagnostic workup of a suspected brainstem glioma should always include metabolic imaging. MR spectroscopy is a potential aid in this setting. However, due to multiple limiting factors of MRS extensive application have not begun.

Up to this day only two retrospective analyses are available with FET PET/CT performed for infratentorial and/or spinal glial tumors both with a relatively low number of scans (36 and 16, respectively) [30,31]. Both studies showed the additional benefit of FET PET/CT, yet their conclusions and discussions are only partially matching.

Since BSG are rarely amenable to biopsy, only a limited number of cases can be histologically verified. Both studies have low number of verified cases (14/36 and 9/16, respectively) [30,31]. For further correlation clinical follow up or MRI scan data were used. There was no significant difference between TBR and SUV measurements in either study in the high grade glioma (HGG) and the low grade glioma (LGG) subgroups. Due to the special circumstances not all diagnostic indexes could be exactly calculated. Therefore, only descriptive analysis could be interpreted. In the Tscherpel and co study 5 out of 11 low grade glioma showed TBRmax greater than 2.5. Also 3 of the 6 verified LGGs showed TBR less than 1.6. However, in this dataset tumor progression could be unequivocally identified with FET PET/CT [31]. The main conclusion of the Albatly and co group was that the higher TBR values are more likely to predict future progression than lower TBR lesions. Their TBR cutoff value for this statistical separation was 2.0 [30]. These partially conflicting findings demonstrate the difficulties of a general application and common understanding the role of FET PET/CT in managing BSG.

Further elaboration should also note the importance of spatial genetic heterogeneity of all glial tumors. Since surgical options are often limited our knowledge about false negative or misrepresentative biopsy sites is scarce [32]. Available single-center and single-surgeon experiences pose the risk of bias in both directions [32,33,34]. A metanalysis of surgical management of BSG also showed also ambivalent findings [35]. The number of fully traceable published series are very limited. All together 213 surgical resection and 125 biopsy cases were evaluated. One of the few general findings was that when surgery was possible it could improve survival however at a very dearly cost (complication rate of biopsy 10%, surgery 35%). Other outspoken statement is the key role of radiotherapy [2]. This could however not be stated for any kind of chemotherapy [1,36,37], which was usually used as last resort. However new data suggest that H3K27M mutant glioma patients can benefit from dordaviprone treatment [38,39].

FET PET/CT is already a recommended, non-invasive modality in decision making in the supratentorial setting [19]. In our study we aimed to test and analyze FET PET features of adult BSG. Since only 3 cases were histologically verified, we relied on follow up clinical and MRI data. Using previously applied TBR cutoff values we found no statistical difference between subgroups. However, after increasing the TBR cutoff to 2.9, we were able to separate two statistically significant subgroups with a high sensitivity (95%). Using the relatively higher TBR we could correctly identify long term survivals with good positive and negative predictive values.

These facts underline the importance and reliability of FET PET/CT in the diagnostic workup of brainstem tumors.

Certain limitations have to be addressed. Only static FET PET/CT scans were available and analysis was based on clinical follow-up data. There was no “gold standard” term of comparison because surgical verification was restricted, therefore false negativity and false positivity could not be ruled out. Furthermore, specificity was undesirably low in this setting. Also, the low number of available patients had a limiting effect on our analysis. In the future PET MRI BSG is a rare disease with high scientific interest. Yet only low number of FET PET data sets are available on BSG. For further advancements in the field multinational multicenter studies are needed. Founding an international brainstem tumor ’bank’ could help reaching the desired number of surgically safe biopsy sites and the histological data. Further development could be reached by widespread use of PET/MRI which is the most optimal hybrid diagnostic modality in order to obtain clear anatomical and metabolic qualities of uncertain brainstem lesions.

Guidelines often highlight the role of FET PET/CT only in the posttreatment setting of supratentorial gliomas. In our experience static FET PET/CT can aid tumor volume identification for infratentorial gliomas as well, both for radiation and surgical planning

In our analysis we were able to highlight its crucial role during the initial diagnosis and prognostic prediction.

## 5. Conclusions

In this study a TBR cutoff value was estimated, yielding a slightly higher cutoff than in previous references [22,24,25,40,41,42]. However this cutoff resulted in a good separation between survival times in the two TBR categories (*p* = 0.026), and in a high sensitivity value (proportion of low TBR values among survivors was 91.6%, 95% CI from 76.0% to 100%). Future aim is to test the long term durability of our proposed TBR, we are planning on performing a 5 year validation.

## Figures and Tables

**Figure 1 cancers-17-03065-f001:**
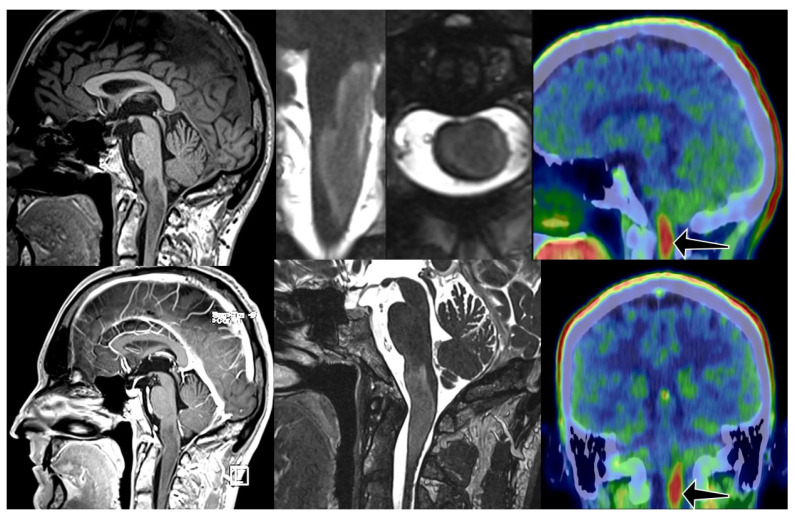
Diffuse intrinsic tumor from medulla oblongata till C 2 level with no enhancement on MRI. FET PET CT showed intense FET uptake in the lesion (arrows). TBR was 2.7. Initial diagnosis was obtained in March 2021. The patient underwent conformal radiotherapy (54/1.8 Gy) and follow up scans showed some regression. Unfortunately, 3 years after irradiation MRI showed slight progression, patient started TMZ chemotherapy. 1236 days after initial diagnosis patient is doing fine with minor symptoms.

**Figure 2 cancers-17-03065-f002:**
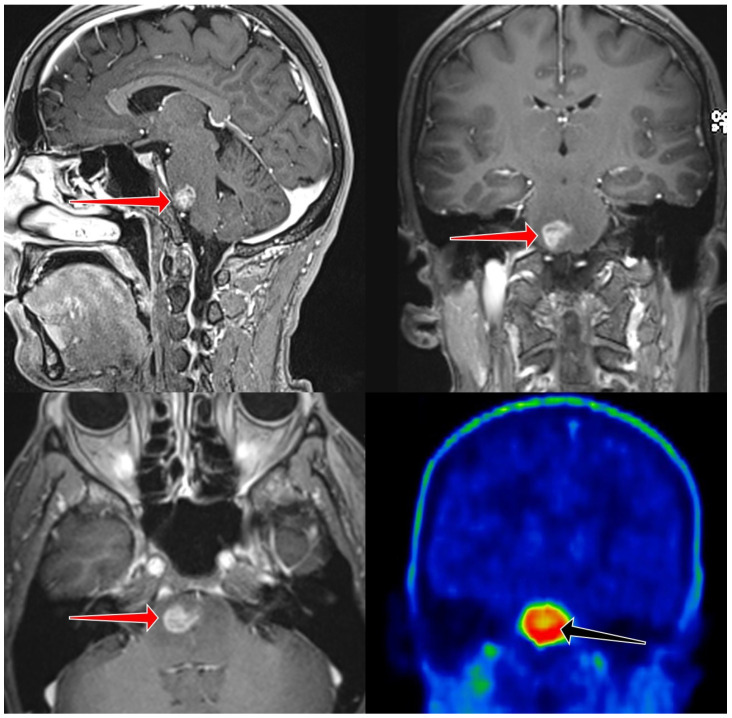
High grade, clinically malignant subtype, with contrast enhancement (red arrows). PET CT also showed intense FET uptake (black arrow), TBR was 3.9. Patient died before therapy, 28 days after MRI scan.

**Figure 3 cancers-17-03065-f003:**
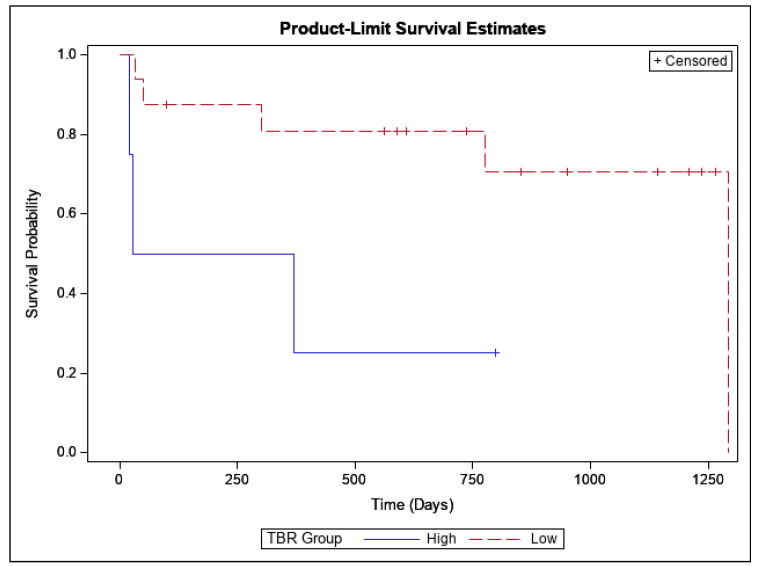
Kaplan-Meier graph of survival distributions.

**Table 1 cancers-17-03065-t001:** Descriptive statistics of TBR values.

Category	N	Mean	S.D.	Min	Median	Max
Death cases	8	2.30	1.00	1.2	2.14	3.9
Survival cases	12	2.07	0.82	1.1	2.10	3.9
Overall	20	2.16	0.88	1.1	2.10	3.9

**Table 2 cancers-17-03065-t002:** Performance characteristics of the cutoff TBR = 2.9.

				95% CI	
Statistic	*n*/N	Estimate (%)	S.E. (%)	Lower Limit (%)	Upper Limit (%)
Sensitivity	11/12	91.7	8.0	76.0	100.0
Specificity	3/8	37.5	17.1	4.0	71.0
Positive Predictive Value	11/16	68.8	11.6	46.0	91.5
Negative Predictive Value	3/4	75.0	21.7	32.6	100.0

## Data Availability

All used data is available for special request.

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
