# Peer review of "Brainstem Glioma Prognostication: Static FET PET/CT"

_cancers, 2025, doi:10.3390/cancers17183065_

Round 1
Reviewer 1 Report
Comments and Suggestions for Authors
Simple summary and abstract: Essentially identical but both adequately summarizing the submission. The simple summary can be shortened.
Introduction: The introduction provides a background to the remainder of the submission, it must take into account up to date neuro pathology. First, on lines 42-43 the authors state “despite all these disadvantages long-term prognosis can not be much better than for hemispherical gliomas. ” This is confusing and requires elaboration as brainstem gliomas typically have a worse prognosis than supratentorial gliomas. No references made to current terminology of midline glioma that is now utilized for tumors with an H3K27M mutation (infratentorial as well as supratentorial and spinal tumors can have this mutation). Biopsy of tumors in the brainstem is uncommon and (though not the goal of this study) functional imaging such as Fluorine-18-labeled ethyl-l-tyrosine (FET) PET-CT may also be helpful in distinguishing between midline gliomas and other pathologic entities as these are treated differently (potential use of temozolomide for gliomas and use of the recently US FDA approved ONC201 for midline gliomas). The authors presented retrospective analysis using FET PET-CT to determine any prognostic aspects utilizing this technology in the evaluation of braiinstem gliomas. Recommendations for the introduction is to clarify the pathology terminology and to change the wording of lines 42-43.
Materials/methods: Straightforward, no specific concerns regarding data retrieval and analysis.
Results: With a cutoff of the tumor to brain ratio of 2.9, a statistically significant difference was seen with regard to a poor prognosis group and a better prognosis group with regard to median overall survival. No specific concerns with regard to methodology utilized in deriving this result.
Discussion: Adequately addressed and interpreted the results an adequately addressed limitations. If possible, I would like to see a sentence or 2 regarding if the authors believe that this would be helpful in helping distinguish midline gliomas from other histologies.
Conclusions: Adequately summarizes the discussion. The conclusion should also mention prospects for further study.
References: All references are up-to-date as to the
Tables and figures: All are supplementary or complimentary to the narrative text and not redundant.
Author Response
Reviewer #1
Thank you very much for taking time to review our article and sharing your expertise and valuable remarks. Please find our detailed responses listed below
- Simple summary and abstract: We shortened the summary.
- Introduction:
Thank you for highlighting the importance of using correct pathological terminology. Our intention was to reflect that although the term brainstem glioma suggests poor prognosis, in significant percentage of cases, survival could be better than glioblastoma. We did not want to elaborate details of pathological entities. Aim was to separate poor and better prognosis groups. We changed the wording to
” Despite all these disadvantages long-term prognosis can be much better than for hemispherical glioblastomas”
Also thank you for pointing out a new official treatment modality for midline gliomas, we added this information to the introduction as well.
“This could however not be stated for any kind of chemotherapy[1,32,33], which was usually used as last resort. However new data suggest that H3K27M mutant midline glioma patient can benefit from dordaviprone treatment.”
- Discussion: If possible, I would like to see a sentence or 2 regarding if the authors believe that this would be helpful in distinguishing midline gliomas from other histologies.
Thank you for the professional interest in our opinion.
Adequate histological verification would be essential to validate metabolic scans, but also metabolis scans are vital to surgical biopsy. To achieve that extensive functional planning (DTI) and further improvement in minimal invasive infratentorial biopsy technics are needed. Our opinion is that maybe an international brainstem tumor ’bank’ would a possible way in reaching the optimal number for verified cases. We expanded the section the following (lines 224-226):
” For further advancements in the field multinational multicenter studies are needed. Founding an international brainstem tumor ’bank’ could help reaching the desired number of surgically safe biopsy sites and the histological data.”
- Conclusions:
Thank you for pointing out the importance of future forward thinking. Our longer-term goals include to validate these patients data with 5 year follow up. We updated the section the following:
”Future aim is to test the long term durability of our proposed TBR, we are planning on performing a 5 year validation”
Reviewer 2 Report
Comments and Suggestions for Authors
Excellent work demonstrating the potential use of PET imaging in the treatment of gliomas. It is worth exploring the role of PET and radiomics in distinguishing true progression from pseudoprogression after brain radiotherapy, significantly improving the oncological management of these diseases. This worthy-of-citation study discusses this in the radiomics session: https://doi.org/10.3390/brainsci12040416.
These tumors are very difficult to treat with chemoradiotherapy and tend to recur inexorably, as can be seen from this worthy-of-citation study on Childhood Progressive Diffuse
Intrinsic Pontine Glioma (DIPG) .
Another aspect that deserves further exploration is the use of H3K27M-Mutant
as a predictive index in this type of Pontine Glioma.
Good work.
Author Response
Reviewer #2
Thank you very much for taking time to review our article and sharing your expertise and valuable remarks. Please find our detailed responses listed below
- Radiomics:
Thank you for pointing out other great study on further prospects of PET CT. We only included radiomics in terms of MRI reading. Radiomics could be the next decisive improvement in ’reading’ metabolic scans as well. It is indeed worth of mentioning as the new potential ancillary evaluation form. We expanded the Materials and methods section the following
” Literature can also be inconsistent on what cutoff values should be used for true validation of malignancy and tumor delineation. In the near future radiomic analysis can be a helpful aid to identify these qualities. Our main goal was to identify the best possible TBR value that could reflect on tumor prognosis.” - Treatment modalities:
Thank you for pointing out the difficulties of chemotherapy. We updated the introduction section the following.
““This could however not be stated for any kind of chemotherapy[1,32,33], which was usually used as last resort. However new data suggest that H3K27M mutant midline glioma patients can benefit from dordaviprone treatment.” - Predictive index:
Thank you for highlighting a very important genetic feature and its vital role in tumor prognosis. We improved the wording in the introduction section (lines 70-71)
“Certain genetic features can be used as prognostic factor such as IDH mutation is for supratentorial gliomas. For brainstem gliomas H3K27M mutation represents a similar role [10,11]”
Reviewer 3 Report
Comments and Suggestions for Authors
This is a very interesting study, and the author should be congratulated for their efforts.
Minor revisions are suggested.
Introduction is very elaborate, it needs to be crisp and concise. Much of it can be moved to discussion section. Also, discuss the clinical implications of identifying risk groups. Should radiation dose-escalation be considered in high-risk group?
Authors should also acknowledge the very low specificity.
Line 55: high grade glioma
Lines 87-90: formatting issues: decimal points were mistakenly replaced by commas in the numbers
There are formatting errors throughout the manuscript, please proof-read once again.
Author Response
Reviewer #3
Thank you very much for the thorough review and useful remarks. Please find our detailed responses listed below
- Introduction: Thank you for your opinion and recommendations. We moved the last 4 paragraphs of the introduction to the discussion section, as advised.
- Discussion: Thank you for pointing out potential further discussion points. Our primary intention was to provide prognostic information.After thorough analysis of the literature and our regional practice we did not want to go in details in therapy recommendations.
- Low specificity: We updated the limitation part of the conclusion section as the following:
“Certain limitations have to be addressed. Only static FET PET/CT scans were available and analysis was based on clinical follow-up data. There was no “gold standard” term of comparison because surgical verification was restricted, therefore false negativity and false positivity could not be ruled out. Furthermore specificity was undesirably low in this setting. Also the low number of available patients had a limiting effect on our analysis”.
Reviewer 4 Report
Comments and Suggestions for Authors
The manuscript conceptualization and design are well thought out. The article could aid in reaching a sound clinical judgement while using F FET PET CT for brain stem gliomas.
Would adding a n MRI PET to FET PET CT improve the specificity?
Author Response
Reviewer #4
Thank you very much for your efforts reading and assessing our work. Please find our detailed responses listed below
- Specificity:
Thank you for raising the issue of PET MRI. In our opinion PET/MRI would obviously increase all stastics markers since metabolic and anatomic qualities would be recorded in a single setting. However, we don’t think that only the co-registration would solve the low or lower specificity problem. In order to provide high specificity unexceptional histological validation would be needed. That could help further understanding and better reading of MRI as well PET/CT images. We consider PET/MRI to be the most optimal hybrid imaging modality with the highest diagnostic accuracy in brain diseases, with particular regard to localization and type of disease.
We updated the discussion section as the following (lines 223-228):
”For further advancements in the field multinational multicenter studies are needed. Founding an international brainstem tumor ’bank’ could help reaching the desired number of surgically safe biopsy sites and the histological data. Additional development could be reached by widespread use of PET/MRI which is the most optimal hybrid diagnostic modality in order to obtain clear anatomical and metabolic qualities of uncertain brainstem lesions.”